# Rare Superior and Middle Trunk Fusion Accompanied by Altered Division Rearrangement Results in a Unique Brachial Plexus Variant: A Case Report

**DOI:** 10.3390/diagnostics14121239

**Published:** 2024-06-12

**Authors:** Andreas Marco Schlüter, Konstantin Redl, Thomas Tschernig, Stephan Maxeiner, Gabriela Krasteva-Christ

**Affiliations:** 1Anatomy and Cell Biology, Saarland University, Kirrbergerstr. 100, Bldg. 61, 66424 Homburg, Germany; ansc00014@stud.uni-saarland.de (A.M.S.); kore00001@stud.uni-saarland.de (K.R.); thomas.tschernig@uks.eu (T.T.); stephan.maxeiner@uni-saarland.de (S.M.); 2Center for Gender-Specific Biology and Medicine (CGMB), Saarland University, 66424 Homburg, Germany

**Keywords:** cadaveric dissection, plexus brachialis, anatomical variation, trunk fusion, case report

## Abstract

During routine dissections of cadavers as part of the medical curriculum, we identified a rare unilateral variation in the brachial plexus on the right side of a female body donor. This variation consisted of four unusual changes to the regular pattering of nerve bundles and the dorsal scapular artery permeating the complex neural network. The variation included contributions of root C4 to the plexus by a root C4/C5 anastomosis, a rare fusion of the superior and middle trunks to a ‘superomiddle’ trunk, a preliminary, proximal branching of the suprascapular nerve off the C5 root. We further observed an accessory ‘medial anterior division’ branching off the fused upper and middle trunks merging with the anterior division of the inferior trunk forming the medial cord. The latter event potentially introduced nerve fibers from C5 to C7, which are absent in common patterns. We aim to relate these observations to previous categorizations and quantifications of brachial plexus patterns. We believe that the combination of different variations in this case resulted in a unique pattern. Since this observation was made in the dissection class, we further aim to raise awareness among medical students and anatomical instructors for the likelihood of variations to textbook patterns. This will hopefully foster an appreciation of uniqueness and individuality in the interaction with future patients demonstrating that proper preparation prior to surgical interventions is always a necessary prerequisite.

## 1. Introduction

The brachial plexus is an important network of nerves extending from the neck to the axilla, surrounding partially the subclavian and axillary arteries. It provides the sole source of somatic motor and sensory innervation of the upper limbs eventually dissolving distally into five separating major nerves, i.e., the musculocutaneous, axillary, radial, median, and ulnar nerve. The brachial plexus is further divided into several structural subregions from which additional nerves branch out that are responsible for the innervation of muscles in the scapular and pectoral regions [1,2].

Most proximally, five roots emerge through the interscalene fissure from a fusion of ventral spinal nerve rami representing spinal cord segments C5 to C8 and T1. The roots of C5 and C6 merge into the superior trunk, C8 and T1 merge into the inferior trunk, whereas C7 generally by itself represents the middle trunk. Contributions of ventral rami of C4 or T2 have also been observed [2]. These contributions might be extradural or intradural [2,3]. A brachial plexus with C4 contribution to the superior trunk is referred to as a prefixed brachial plexus; contributions by T2 to the inferior trunk are consequently referred to as postfixed [2]. In regular arrangements of the brachial plexus, all three trunks separate into a total of six divisions, an anterior and posterior division of each trunk, respectively [1,2]. These divisions subsequently rearrange into three cords. The cords are distinguished according to their relative position to the axillary artery. The anterior divisions of the superior and middle trunk combine into the lateral cord, all posterior divisions merge into the posterior cord, and the anterior division of the inferior trunk proceeds as the medial cord. Eventually, the lateral and medial cords branch out into the musculocutaneous and ulnar nerves, respectively; separate branches (the lateral and medial ‘heads’) of both cords, however, jointly merge into what becomes the median nerve. The posterior cord branches out into the axillary and radial nerves [1,2].

Variations in the brachial plexus are common and have already been reported for over a century [2,4,5]. Whereas numerous reports focus on the frequency of single variations in nerves or blood vessels, less has been published summarizing and classifying these changes together with the respective frequencies of these variations [2,4,6]. Variations in the brachial plexus are considered to result from changes to the sorting process of nerves innervating the upper limb/shoulder during embryonic development [2]. These changes are due to the origin of the nerve, the distribution of the target of the nerve, and the relationship with adjacent structures [2]. In particular, the origin and course of the subclavian and axillary arteries are impacting the formation of the brachial plexus [2,7,8]. Already in 1918, Arthur Kerr catalogued 38 distinct patterns based on the dissection of 175 brachial plexuses [4]. Most recently, Benes and colleagues presented a meta-study incorporating data from over three thousand upper limbs (40 studies) and categorized different rearrangements, including their respective frequencies [6]. For roots and trunks, the authors classified root shifts, variations in root numbers, i.e., incorporation of C4 or T2, a combination of both, variations in trunks resulting from those roots (two or four trunks), and a combination of numbers of trunks and roots. They observed that 84% of all roots and trunks follow the regular pattern, and that the only two other noteworthy variants were the incorporation of C4 to the superior trunk (7% of cases) and T2 to the inferior trunk (0.2% of cases). The contribution of C4 or T2 warranting a conclusive assessment of pre- or postfixed cases varies in the literature [9]. The exchange of small anastomoses between C4 and C5, however, has been reported to be relatively frequent (41%) [10]. Variants regarding the divisions and cords do not account for more than 4% of all variants, including shifts, numbers of divisions, or quantity of cords [6].

General awareness of the variations in the brachial plexus and arteries in this region is of utmost importance for neurologists and orthopedists, or for anesthetists, radiologists, and neurosurgeons planning any kind of surgical procedures or interventions. 

This case report highlights five differences to the most common pattern of the brachial plexus. We present differences in trunk and cord formation, the occurrence of an additional anterior division to the medial cord, and the re-routing of the dorsal scapular artery. Some of these variants are more frequently reported than others. The combination of all variants together, however, leads us to conclude that this case report presents a very unique pattern of the brachial plexus.

## 2. Case Presentation

Students of the third semester of medical studies at Saarland University routinely dissect formalin-fixed cadavers of body donors aiming at the presentation of various anatomical structures such as the brachial plexus. In a 77-year-old female Caucasian body donor, we observed a unilateral variation in the brachial plexus on the right side, whereas a regular arrangement of the plexus was observed on the left side (Figure 1). We identified five differences (see description below) to the regular arrangement at various levels, respectively, at the roots, roots merging into trunks, and divisions separating from trunks rearranging into cords. 

At the level of the roots, the C4 root made an extradural contribution to the brachial plexus, sending nerve bundles toward the C5 root prior to the formation of the superior trunk (Figure 1c) reminiscent of a prefixed-like brachial plexus in which contributions by C4 are included. Some extradural bundles of the C5 root likewise split off and join the C4 root. The suprascapular nerve (SSN) originated proximally from the superior trunk of the brachial plexus than in the typical variant. On the level of trunk formation, C4, C5, C6, and C7 roots fused into one common trunk instead of forming two separate trunks, i.e., a superior (ST, including C5/C6 contributions) and middle trunk (MT, including C7 contribution) (Figure 1b). We have proposed a name, ‘superomiddle trunk’ (SMT), for this structure. The inferior trunk (IT) appears unaltered with sole contributions by roots from C8 and T1. The presence of only two trunks consequently resulted in a rearrangement of five instead of six divisions. The SMT gave rise to a larger lateral anterior division (lADSMT) similar to the anterior division of the ST in general patterns of the brachial plexus (Figure 1a) forming the lateral cord, and an additional smaller medial anterior division (mADSMT). The mADSMT merged with the anterior division of the inferior trunk forming the medial cord (Figure 1b,c). Posterior divisions (PDs) of both SMT and IT merged into the posterior cord. Further, distally, the formation of the five major branches innervating the upper limb was indistinguishable from the general pattern of the brachial plexus. The variations are summarized in the schematics presented in Figure 2. Finally, the formation of the ‘superomiddle trunk’ necessitated an alternative route for the dorsal scapular artery (DSA), which has more frequently been reported to descend between the superior and middle trunk into the direction to the shoulder [2,11]. In this case report, the presence of a superomiddle trunk naturally impedes the course of the DSA between the upper and middle trunk. Here, the DSA descended caudal of the SMT and proximal to the mADSMT and PDIT (Figure 2b).

## 3. Discussion

Variations in the regular pattern of brachial plexuses have been documented in the literature, starting about a century ago, with seminal observations by Arthur Kerr [4]. Given the literature, in particular, the categorization and quantifications presented in a most recent meta-study, we aimed to demonstrate that the combination of variants in our case report led to a quite unique pattern of a brachial plexus [6]. 

At the level of the roots, we observed a prefixed-like extradural C4 and C5 anastomosis, which commonly appear in over 22–63% of cases, with intradural connections in 17–36% of cases [2]. Benes et al. reported a total of 11% of cases [6]. The estimation of C4 contributions varies in the literature, and the term prefixed is not strictly defined [2,3]. Some authors justify this term with a major contribution to the plexus [12], and others note major contributions to the superior trunk by C4 and major contributions of C8 to the inferior trunk [13]. In our case, we cannot entirely rule out that nerve fibers from the C4 root may end up distally in all five major nerves. The chance of this occurring, however, is unlikely. In a recent study, C4 contribution in prefixed brachial plexuses to different nerves has been ranked [14]. Most frequently, these fibers end in the suprascapular nerve, the musculocutaneous nerve, and axillary nerve. Despite the observation that the C4 contribution to the C5 root in our case is considerably small, the likelihood of C4 fibers contributing to the suprascapular nerve (SSN) may be high since the SSN branches off the C5 root prior to trunk formation, leaving out a contribution by C6. Generally, C5 and C6 contributions are considered to be present in the SSN, which is either branching off more distally from the upper trunk or its posterior division to the posterior cord [2]. The lack of C6 fibers may potentially be substituted by C4 fibers. 

Our case displays an unusual fusion of the C5 (with the potential inclusion of C4 fibers), C6, and C7 roots to what we would suggest calling a superomiddle trunk instead of a separate upper and middle trunk. We can rule out that this is a dissection artifact due to the incomplete removal of, e.g., connective tissue, which might obscure the presence of separate trunks. Trunk fusions occur but are incredibly rare. Upper/middle trunk fusions or middle/lower trunk fusions represent less than 0.1% of all variations in trunk numbers, respectively [6]. Higher frequencies for an upper and middle trunk fusion ranging from 2% to 4% were reported elsewhere [4,15]. Unilateral fusion of the superior trunk with the middle trunk under the contribution of C5, C6, and C7, without further specifying any additional variations in the dissected plexus, has been reported previously [16,17]. In contrast to the formation of upper and middle trunk fusions, cases of fusions of the middle trunk and the inferior trunk have also been reported [18,19,20]. 

Whereas the (lateral) anterior division of a fused upper and middle trunk might be interpreted as a contraction of the respective anterior divisions arising from the middle trunk and upper trunk in a regular pattern of a brachial plexus, the presence of an additional medial branch of the superomiddle trunk, i.e., a medial anterior division forming with the anterior division of the inferior trunk the medial cord, is rather unexpected. This formation potentially introduced fibers of roots C5, C6, and C7 to the medial cord and its branching nerves, which normally do not receive any fibers from these roots, such as the ulnar nerve, or the medial brachial and antebrachial cutaneous nerves. This is a situation non-existent in any regular arrangement. What might be the reason to add an accessory medial anterior division to the superomiddle trunk? To address this question with a plausible answer, we need to meet two prerequisites, the assumption that eventually the destinations of all nerves remain the same when redistributed through variations in plexus patterns and that obstructions during development might account for these changes in plexus patterns [2,7,8]. If we focus primarily on the five distal nerves leaving the plexus, the only major nerve that necessitates an additional anterior division fusing with the medial cord could be nerve bundles from C7 leading up to the median nerve now “re-routing” through the medial cord and the medial head of the median nerve rather than exclusively through the lateral cord and its lateral head. Knowledge of alternate nerve routes is important prior to surgical interventions [21]. Since this alternative route does not provide C7 fibers to the musculocutaneous nerve, we are tempted to speculate that during development, the C7 fiber bundles needed to split up once leaving the superomiddle trunk. This observation could possibly have implications in, e.g., accidents related to ruptures of the medial cord possibly affecting more severely the upper limb functions related to the innervations provided by the median nerve.

A final difference adding to the unique variation in our case report to common plexus variations was not in regard to the pattern formation but the course of the dorsal scapular artery. DSA might arise from the subclavian artery or the thyrocervical trunk. In the latter case, the DSA does not pierce the brachial plexus [2]. However, when it originates from the subclavian artery, it pierces more frequently between the upper and middle trunk (46–69%) and less frequently between the middle and inferior trunk (18–44%). In a recent study of DSA relationship with the brachial plexus, the course of the DSA arising from the subclavian artery was slightly in favor of a piercing between the middle and inferior trunk [22]. 

Analyses of brachial plexus variations suggest that obstructions during early plexus formation, i.e., axon growth into the budding limbs during embryogenesis by the subclavian artery enforces changes to the more general pattern [7,8]. This may also affect the re-routing of the smaller dorsal scapular artery in our case between the fused superior and middle trunk and the inferior trunk, suggesting that the budding of the DSA occurs after the formation of the trunks.

## 4. Conclusions

Knowledge of the textbook pattern of the brachial plexus and potential variants is important for any kind of planned interventions or surgical procedures affecting the upper limbs, the axilla, or shoulder. Additional contributions by root C4, the presence of a superomiddle trunk, its additional anterior division to the medial cord with the potential contribution of C5, C6, and C7 to nerves branching off the medial cord, as well as a rerouting of the dorsal scapular artery makes this brachial plexus in all its accumulated variations quite unique.

## Figures and Tables

**Figure 1 diagnostics-14-01239-f001:**
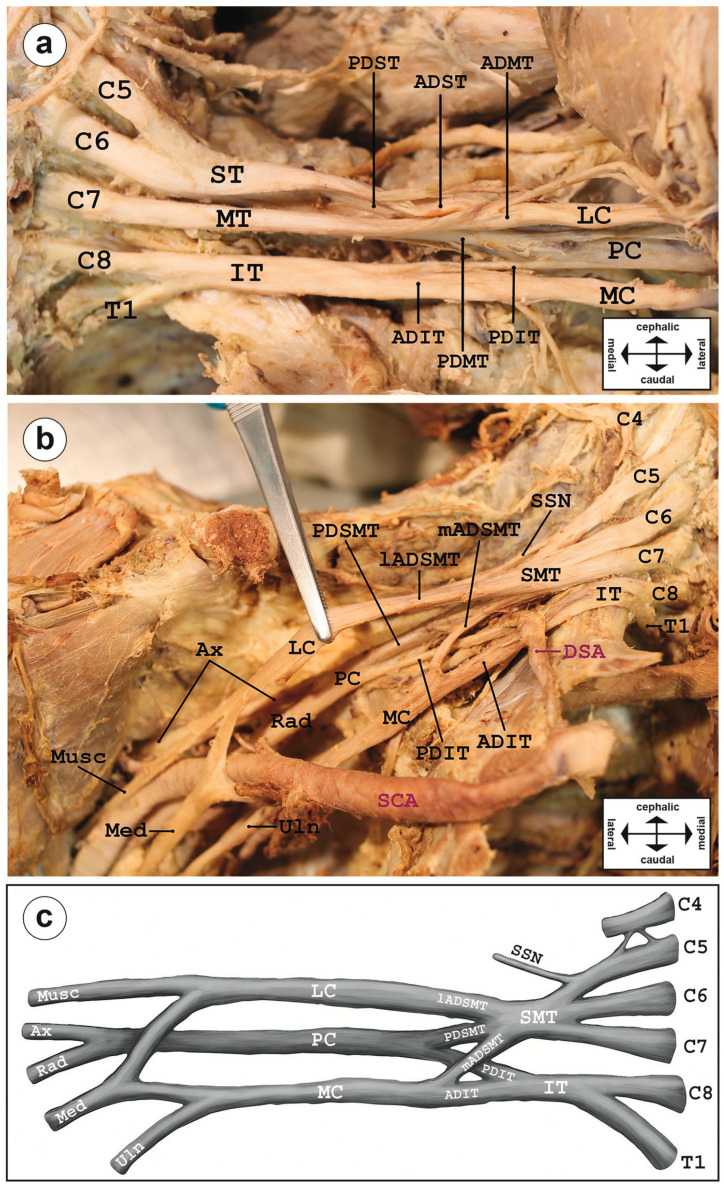
Variation in the brachial plexus. (**a**) Regular formation of the brachial plexus in a cadaveric specimen was observed on the left side of the body. Superior and inferior trunks resulted from fusions of roots C5 and C6 as well as C8 and T1, respectively. Fibers of all three trunks redistributed in anterior and posterior divisions, and, eventually, in lateral, medial, and posterior cords prior to exiting the axilla into five main branches, i.e., the musculocutaneous, axillary, median, radial, and ulnar nerve, respectively. (**b**) Variation in the brachial plexus in situ of the right body side, and a sketch thereof focusing on its main features (**c**). Segments C5 to C7 fuse to a single ‘superomiddle trunk’ (SMT), leaving C8 and T1 to form the inferior trunk (IT). The lateral anterior division of the SMT (lADSMT) solely formed the lateral cord; fiber tracts leaving the SMT as a medial ADSMT together with the anterior divisions of the IT (mADSMT and ADIT) merged into a joint medial cord. Abbreviations: C4 to C8 and T1, spinal cord segments; IT, inferior trunk; MT, middle trunk; ST, superior trunk; SMT, superomiddle trunk; AD, anterior division; ADST, anterior division of the superior trunk; lADSMT/mADSMT, lateral/medial anterior divisions of the superomiddle trunk; ADIT, anterior division of the inferior trunk; PD, posterior division; PDST, posterior division of the superior trunk; PDSMT, posterior division of the superomiddle trunk; PDIT, posterior division of the inferior trunk; LC, lateral cord; MC, medial cord; PC, posterior cord; Ax, axillary nerve; Med, median nerve; Musc, musculocutaneous nerve; Rad, radial nerve; Uln, ulnar nerve; DSA, dorsal scapular artery; SCA, subclavian artery.

**Figure 2 diagnostics-14-01239-f002:**
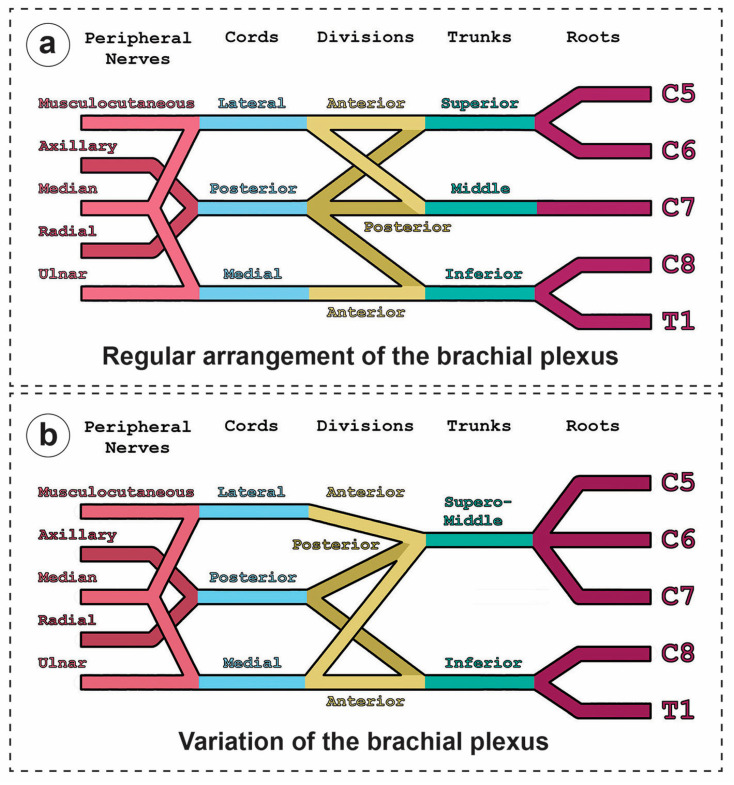
Schematic of the brachial plexus variation. (**a**) General (‘textbook’) pattern of the brachial plexus and (**b**) variation found on the right side of a female body donor. Notable differences were evident in the fusion of the spinal cord segments C5, C6, and C7 to a joint superomiddle trunk (SMT). At its distal end, the SMT branches off into a lateral anterior division exclusively comprising the lateral cord, a posterior division fusing with the posterior division of the IT forming the posterior cord, and an additional medial anterior division merging with the anterior division of the inferior trunk jointly forming the medial cord.

## Data Availability

Data can be obtained from the corresponding author upon reasonable request.

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
