# Peer review of "Rare Superior and Middle Trunk Fusion Accompanied by Altered Division Rearrangement Results in a Unique Brachial Plexus Variant: A Case Report"

_diagnostics, 2024, doi:10.3390/diagnostics14121239_

Round 1

Reviewer 1 Report

Comments and Suggestions for Authors

The report submitted for my review requires significant corrections. The style needs to be improved. Avoiding self-invented terminology is essential, as it can confuse the reader and detract from the report's credibility. The authors use such terminology to describe the observed variation. This approach introduces information noise.

Specific comments:

- The sentence "We identified five major differences (see below) to any regular arrangement at different stages starting at the roots, roots merging into trunks, and divisions separating from trunks rearranging into cords" should be changed to: "We identified five differences (see description below) to the regular arrangement at various levels, respectively at the roots, roots merging into trunks, and divisions separating from trunks rearranging into cords."

- "At the stage of roots" must be replaced by "At the level of roots."

- "C4 root contributes nerve fibers toward the C5 root prior to the formation of the trunk" must be replaced by "C4 root contributed to the brachial plexus sending nerve bundles toward the C5 root prior to the formation of the superior trunk." Nerve fibers are axons that form nerve bundles.

- The sentence "The suprascapular nerve (SSN) branches out proximally but not distally from what is generally the superior trunk" could be clarified for better understanding. For example, "t. The suprascapular nerve (SSN) originated more proximally from the superior trunk of the brachial plexus than in the typical variant."

- The sentence "On the level of the trunk formation, C5, C6 and C7 form a 'superiomiddle trunk' (SMT), instead of being separate entities in the form of a superior trunk (ST, including C5/C6) and middle trunk (MT, including C7) (Fig. 1b)" must be rewritten to avoid usage of self-created terminology out of Terminologia anatomica, e.g., "On the level of the trunks formation, C4, C5, C6 and C7 roots fused into one common trunk, instead of forming two separate superior (ST, including C5/C6 contributions) and middle trunk (MT, including C7 contribution) (Fig. 1b). 

We have proposed a name, 'superomiddle trunk' (SMT), for this structure." Please note that the term "superomiddle trunk (SMT)" is better than "superiomiddle trunk (SMT)."

- The authors describe the C4 contribution to the trunk formation but consistently omit this contribution in further descriptions.

The discussion is very superficial. In the discussion, the authors should refer to more extensive literature. It is crucial to emphasize that variations of blood vessels may accompany variations of nerves. Please refer to Bergmann's encyclopedia of anatomical variations: https://www.anatomyatlases.org/AnatomicVariants/NervousSystem/Text/BrachialPlexus.shtml and the latest guidelines on reporting of anatomical variations: https://doi.org/10.1016/j. trio.2024.100284. Both sources mentioned should be cited.

Comments on the Quality of English Language

Language must be improved.

Author Response

We would like to thank Reviewer 1 for his/her critical comments on our manuscript. We aimed at addressing the issues that have been raised. Please, refer to the sections below:

Comments and Suggestions for Authors

The report submitted for my review requires significant corrections. The style needs to be improved. Avoiding self-invented terminology is essential, as it can confuse the reader and detract from the report's credibility. The authors use such terminology to describe the observed variation. This approach introduces information noise.

We are grateful for this comment and changed the manuscript accordingly. We toned down the use of the term “superomiddle trunk” and picked up the suggestion to change the originally used term “superiomiddle trunk” to “superomiddle trunk”.

Specific comments:

- The sentence "We identified five major differences (see below) to any regular arrangement at different stages starting at the roots, roots merging into trunks, and divisions separating from trunks rearranging into cords" should be changed to: "We identified five differences (see description below) to the regular arrangement at various levels, respectively at the roots, roots merging into trunks, and divisions separating from trunks rearranging into cords."

The wording has been changed accordingly.

- "At the stage of roots" must be replaced by "At the level of roots."

The wording has been changed accordingly.

- "C4 root contributes nerve fibers toward the C5 root prior to the formation of the trunk" must be replaced by "C4 root contributed to the brachial plexus sending nerve bundles toward the C5 root prior to the formation of the superior trunk." Nerve fibers are axons that form nerve bundles.

The wording has been changed accordingly.

- The sentence "The suprascapular nerve (SSN) branches out proximally but not distally from what is generally the superior trunk" could be clarified for better understanding. For example, "t. The suprascapular nerve (SSN) originated more proximally from the superior trunk of the brachial plexus than in the typical variant."

The wording has been changed accordingly.

- The sentence "On the level of the trunk formation, C5, C6 and C7 form a 'superiomiddle trunk' (SMT), instead of being separate entities in the form of a superior trunk (ST, including C5/C6) and middle trunk (MT, including C7) (Fig. 1b)" must be rewritten to avoid usage of self-created terminology out of Terminologia anatomica, e.g., "On the level of the trunks formation, C4, C5, C6 and C7 roots fused into one common trunk, instead of forming two separate superior (ST, including C5/C6 contributions) and middle trunk (MT, including C7 contribution) (Fig. 1b). 

The wording has been changed accordingly.

We have proposed a name, 'superomiddle trunk' (SMT), for this structure." Please note that the term "superomiddle trunk (SMT)" is better than "superiomiddle trunk (SMT)."

The term “superomiddle trunk” has been adopted throughout the manuscript.

- The authors describe the C4 contribution to the trunk formation but consistently omit this contribution in further descriptions.

We specified the potential contribution of C4 in the discussion and included additional references.

The discussion is very superficial. In the discussion, the authors should refer to more extensive literature. It is crucial to emphasize that variations of blood vessels may accompany variations of nerves. Please refer to Bergmann's encyclopedia of anatomical variations:

We appreciate the recommendation of Bergman’s encyclopedia (it’s a great book!) and the web address below. Both have been unbeknownst to the authors. We included the reference and the information into both, introduction and discussion. We further extended the discussion section substantially and separated it into several paragraphs with the aim to add clarity to our train of thoughts.

We removed one section of the discussion describing the classifications by Benes and co-workers and placed it into the introduction.

https://www.anatomyatlases.org/AnatomicVariants/NervousSystem/Text/BrachialPlexus.shtml and the latest guidelines on reporting of anatomical variations: https://doi.org/10.1016/j. trio.2024.100284. Both sources mentioned should be cited.

Comments on the Quality of English Language

Language must be improved.

We have implemented suggestions by Reviewer 2 who is a native speaker and re-assessed the manuscript regarding the improvement of the English language. We aimed at using past and present tenses more coherently and accurately. We hope this has been addressed sufficiently.

We appreciate your time and wish you a good day!

Reviewer 2 Report

Comments and Suggestions for Authors

This article presents a very unique and interesting brachial plexus variation that has multiple simultaneous variations occurring and impacting the dorsal scapular artery and its path to the back of the individual. I think additional references should be included in the introduction to bolster the information being presented there. The figures included with this article are excellent and really helpful in understanding the variations included. I also think the author's explanation for how this variation could arise makes perfect sense. I think stating unique in the title and in the body of the article, rather than "one in a million" is sufficient for describing the distinctiveness of this brachial plexus variation.

Comments on the Quality of English Language

The quality of the English language presented is excellent, but I do include some grammatical changes that need to be addressed.

Author Response

We would like to thank Reviewer 2 for his/her critical comments on our manuscript. We aimed at addressing the issues that have been raised. Please, refer to the sections below:

Comments and Suggestions for Authors

This article presents a very unique and interesting brachial plexus variation that has multiple simultaneous variations occurring and impacting the dorsal scapular artery and its path to the back of the individual. I think additional references should be included in the introduction to bolster the information being presented there.

We thank the reviewer for pointing out this flaw and, accordingly, have added references in our introduction.

The figures included with this article are excellent and really helpful in understanding the variations included. I also think the author's explanation for how this variation could arise makes perfect sense. I think stating unique in the title and in the body of the article, rather than "one in a million" is sufficient for describing the distinctiveness of this brachial plexus variation.

We have toned down our description and adopted the adjective “unique” according to the reviewer’s suggestion.

Comments on the Quality of English Language

The quality of the English language presented is excellent, but I do include some grammatical changes that need to be addressed.

We are very grateful for the kindness of the reviewer to suggest changes to our grammar and spelling. All corrections have been addressed in this resubmission.

We need to point out, however, that substantial revision of the introduction and the discussion has been made as has been suggested by two other reviewers. Where applicable we tried to adhere to the CARE guidelines for the presentation of case reports as suggested by Reviewer 4.

We appreciate your time and wish you a good day!

Reviewer 3 Report

Comments and Suggestions for Authors

The authors present a rare variation of the brachial plexus. It is interesting but the variations of the plexuses are very frequent and numerous.

It is not clear why this variation is "one in a million", what evidence they present and in comparison with the bibliography presented. Therefore the authors should remove the "one in a million" from the title and discussion. In addition, the authors must include very important references such as Tubbs, RD, Mohammadall MS, Loukas M, (2016). Bergman's comprehensive encyclopedia of human anatomic variation

Author Response

Comments and Suggestions for Authors

The authors present a rare variation of the brachial plexus. It is interesting but the variations of the plexuses are very frequent and numerous.

We would like to thank the reviewer to give us the possibility to address his/her concerns. We have observed four differences in the arrangement of the brachial plexus in our case report that deviate from what is more generally assumed to be a “regular” arrangement. Additionally, we point out an alternative route of the dorsal scapular artery. We agree that some of the differences appear with certain frequencies, the statement “one in a million”, however, has been chosen to make a statement about the rarity of all differences together. We have toned down this statement and changed it to unique. The uniqueness of the brachial plexus is rooted in the combination of its less frequent variations.

It is not clear why this variation is "one in a million", what evidence they present and in comparison with the bibliography presented. Therefore the authors should remove the "one in a million" from the title and discussion. In addition, the authors must include very important references such as Tubbs, RD, Mohammadall MS, Loukas M, (2016). Bergman's comprehensive encyclopedia of human anatomic variation

We are grateful for the recommended book by Tubb’s et al. (it’s a great book!) and went through the sections regarding variations of the brachial plexus. This book was of great help and guided us in judging the frequencies that have been reported previously. Additionally, our observation of different frequencies is depending on the cited meta-study by Benes and coworkers.

We rearranged the introduction as well as the discussion sections following now closer the CARE guidelines for case reports. The discussion section has been substantially rewritten and extended, paragraphs have been introduced to add clarity to our train of thoughts (further demanded by Reviewer 4). We hope that these changes meet the criticism of Reviewer 3 and helped to improve the manuscript.

We appreciate your time and wish you a good day!

Reviewer 4 Report

Comments and Suggestions for Authors

This study treats an interesting topic regarding to the brachial plexus and its variants, resulting potentially crucial in this specific field of knowledge. However, the structure of the article appears not well organized and different parts of this case report should be improved in order to consider the study helpful to the research.

Abstract: the abstract should be written in a more scientific way and highlight the aim of the study, in particular the importance of the case reported by the authors to the knowledge of anatomical variants of the brachial plexus.

The Introduction should be more consistent: after describing the usual anatomy of the brachial plexus, it would be useful to specify the different variants, describing the anatomical particularity and the incidence, citing the classification and describing the results emerged from the most recent meta-analysis which supported this topic.

The aim of the study is not well described so that the readers should not identify the key to understanding the study.

The description of the case is confusing: I suggest to organize it in subparagraphs in order to divide the different anatomical variations and attached the relative images, and use the CARE guidelines.

Discussion: in this section the authors did not really discuss the results obtained, in particular it appears as a simple repetition of the case above described. The authors should provide more speculations: for example, the discussion should provide more information about each variants detected in the case report and their relation to the clinical and surgical implications which could influence the diagnosis and treatments of pathologies which involve the brachial plexus.

Furthermore, due to the lack of a clear aim of the study, the authors did not give a real identity to the study itself and not provide a consistent description which does not allow to emphasize the importance of the treated topic, in order to highlight the real contribution to the research of this case report.

Author Response

We would like to thank Reviewer 4 for his/her critical comments on our manuscript. We aimed at addressing the issues that have been raised. Please, refer to the sections below:

Comments and Suggestions for Authors

This study treats an interesting topic regarding to the brachial plexus and its variants, resulting potentially crucial in this specific field of knowledge. However, the structure of the article appears not well organized and different parts of this case report should be improved in order to consider the study helpful to the research.

Abstract: the abstract should be written in a more scientific way and highlight the aim of the study, in particular the importance of the case reported by the authors to the knowledge of anatomical variants of the brachial plexus.

We addressed the aim in the abstract.

The Introduction should be more consistent: after describing the usual anatomy of the brachial plexus, it would be useful to specify the different variants, describing the anatomical particularity and the incidence, citing the classification and describing the results emerged from the most recent meta-analysis which supported this topic.

In the original submission we placed the results of the meta-study in our discussion section. We have moved this section into the introduction.

The aim of the study is not well described so that the readers should not identify the key to understanding the study.

The aim was mentioned in the Abstract.

The description of the case is confusing: I suggest to organize it in subparagraphs in order to divide the different anatomical variations and attached the relative images, and use the CARE guidelines.

This is the very first case report of the authors involved in writing the manuscript. We appreciate the recommendation of the reviewer and consulted the CARE guidelines. In the CARE checklist we found some sections applicable and implemented them accordingly. Other sections regarding patient information, history, therapeutic intervention, follow-up and outcomes, and patient perspective were not applicable.

We believe that the demonstration of anatomical variations as a result from the dissection hall does not meet all of the criteria of CARE.

Discussion: in this section the authors did not really discuss the results obtained, in particular it appears as a simple repetition of the case above described. The authors should provide more speculations: for example, the discussion should provide more information about each variant detected in the case report and their relation to the clinical and surgical implications, which could influence the diagnosis and treatments of pathologies which involve the brachial plexus.

The discussion has been extended substantially, now addressing the variants that have been observed in each paragraph. We also introduced some speculations. The authors do work in an anatomy department, but they are not physicians with experience in surgery. We apologies for reduced references to potential clinical considerations prior to any sort of interventions.

Furthermore, due to the lack of a clear aim of the study, the authors did not give a real identity to the study itself and not provide a consistent description which does not allow to emphasize the importance of the treated topic, in order to highlight the real contribution to the research of this case report.

We do understand the concern of the reviewer. Our intention was to present an interesting observation that was dissected by some interested students from a previous dissection class. I hope that we could address most of the concerns by additions to the introduction as well as the substantial additions to the discussion. We tried to our best to incorporate the CARE guidelines where applicable.

We appreciate your time and wish you a good day!

Round 2

Reviewer 1 Report

Comments and Suggestions for Authors

The manuscript has been significantly improved, and the authors have made all essential corrections. I have no further comments.

Comments on the Quality of English Language

None.

Reviewer 4 Report

Comments and Suggestions for Authors

Dear Authors, 

I appreciate Your improvements. However I cannot reccomend the publication of the manuscript in the present form. 

I suggest You and also the editor, to rewrite and resubmitt the manuscript in another and more convincing form. 

I can suggest You to rewrite totally the abstract. still it is not convincing to me. and by the way the aim must be indicated also in the text not only in the abstract.

I suggest you to improve the introduction, not only in adding more information, which were required, but clearly stating why we should read your manuscript, why it is so worthy to be published and to be read. What does it teach to us. 

I understand this variation occurrence is rare and finding it and documenting it can lead to enthusiasm. 

But, there is a but, i ask to myself, is this variation important for anatomy science? for education purposes (for example, I am wondering how the person who was supervising the dissection reacted to this and managed the students) for clinical sciences? or all?

You have stated it in the discussion, but in this current form is difficult to understand. Maybe splitting this information in dedicated paragraphs and writing in a more plain and traditional style would help. 

I know it, when I started to write my firsts manuscript I liked also to change the style and I found original the way I used to present it. However, You have also keep in Your mind worldwide people is reading your stuff. The main aim to publish anything is to disseminate knowledge...world wide so that everybody can understand it and can take information.  

That's why sometimes articles are very boring. You might have noticed they are all in the same style. but only in this way everyone will be able to read and to get the information and use it for further researches and progresses. 

The CARE question. yes CARE does not apply to all (as PRISMA, as CONSORT etc.) but it is always a guideline which helps you into the writing pathway. and when You use it, you must state it, as well as cite the reference. 

so after so long. I strongly recommend a reject and resubmission, rewriting the article in a more traditional style, making everyone understand and being enthusiastic as you were wen you have found the variation.